# The Be-Home Kids Program: An Integrated Approach for Delivering Behavioral Therapies to Adolescents with Episodic and Chronic Migraine

**DOI:** 10.3390/brainsci13040699

**Published:** 2023-04-21

**Authors:** Licia Grazzi, Danilo Antonio Montisano, Alberto Raggi, Paul Rizzoli

**Affiliations:** 1SC Neuroalgologia-Centro Cefalee, Fondazione IRCCS Istituto Neurologico Carlo Besta, 20133 Milan, Italy; 2SC Neurologia Salute Pubblica e Disabilità, Fondazione IRCCS Istituto Neurologico Carlo Besta, 20133 Milan, Italy; 3John Graham Headache Center, Brigham & Women’s Faulkner Hospital, Harvard Medical School, Boston, MA 02115, USA

**Keywords:** episodic migraine, chronic migraine, adolescents, preventive treatments, behavioral approaches, mindfulness, web-based

## Abstract

Migraine disorders are common in populations of children and adolescents. There are different pharmacological treatments for migraine in young patients, but none have specific indications, and doubts about their efficacy exist. The feasibility and effectiveness of behavioral approaches have already been documented in clinical experiences, and they are generally associated with fewer or no unpleasant effects. Among them, mindfulness practice offers a suitable alternative to other adolescent treatments. We present the results of a pilot study, the Be-Home Kids program, performed during the COVID-19 emergency. It was delivered by web and included education on drug use, lifestyle issues, and six sessions of mindfulness-based behavioral practice. We assessed headache frequency, medication intake, and other psychological variables and followed twenty-one adolescents with chronic or high-frequency episodic migraine without aura for 12 months. Results indicated an overall clinical improvement, particularly a 64% reduction in headache frequency over 12 months. In conclusion, our results indicate that a combined treatment which includes patients’ education and six sessions of mindfulness-based practice delivered over the web, can be of great support in reducing headache frequency, medication intake, and the associated psychological burden disability in adolescent migraine patients.

## 1. Introduction

Migraine is a ubiquitous neurological disorder in the world population, afflicting approximately 1 billion people worldwide, with a clear predilection for the female sex. According to the 2019 Global Burden of Disease Study, migraine is the second leading cause of disability and is the leading cause of disability among all neurological conditions combined. It is also one of the first causes of global loss of healthy life (expressed as disability-adjusted life years [DALYs]) [1,2,3]. Migraine is also prevalent among children and adolescents, with a prevalence of around 7.7% that increases with age, from 3% in young children (3–7 years) to 8–23% in adolescents (11–18 years). The prevalence is slightly higher in boys than in girls before puberty, but both incidence and prevalence increase rapidly in girls from the age of 11 years. Adolescents with migraine suffer from high functional impairment in social and family relationships, school performance, and extracurricular activities [4]. In particular, in this specific population, migraine (counted with tension-type headache) accounted for 37.5% of all-cause prevalence and 7% of all-cause YLDs (Years Lived with a Disability) at the global level, the latter mostly referred to the female groups [5].

There are several preventive pharmacological treatments for migraine in young patients, but none of them have a specific indication for this specific patient population, and legitimate doubts about their efficacy have been raised. Locher’s work, a review with meta-analysis of the literature, evaluated the efficacy and safety of several pharmaceutical classes (β-blockers, anticonvulsants, antidepressants, and antihistaminic and calcium channel blockers to natural supplements) versus placebo, showing that none of the drugs showed a clear and significant reduction in migraine days in the long term compared with placebo [4]. Despite no differences being found with a placebo also with respect to adverse events, it is clear that in real-life situations, patients receiving any long-term treatment are exposed to the risk of developing some kind of side effect. As concluded by Locher and colleagues, the benefits of prophylactic medications should be carefully weighed against their potential harms [4].

The results of Locher’s network meta-analysis support the results of the Childhood and Adolescent Migraine Prevention (CHAMP) study, which found no significant difference between topiramate, amitriptyline, and placebo in reducing migraine days in children. It should be specified that there are pieces of several indirect evidence that the placebo effect is more pronounced in children and adolescents than in adults [6] and, thus, trials are needed that allow quantifying the placebo effect in pediatric migraine through innovative treatment strategies that harness the placebo effect in the treatment of this specific population [5]. There is, therefore, little evidence to support the efficacy of prophylactic treatment in the pediatric population. In this target patient population, the space of non-pharmacological, including non-invasive neurostimulation, nutraceuticals, and behavioral treatments, opens up powerfully [7,8].

Among non-pharmacological treatments, the use of behavioral therapies in the adult population already has been, albeit with a limited degree of evidence of some indications [9,10], and little experiences have been collected in an adolescent population [11].

The feasibility and effectiveness of behavioral approaches have already been documented in clinical experiences of the last decades. These approaches have been shown to be effective in terms of reduction in headache frequency between 36% and 72%, and they are generally associated with fewer or no unpleasant effects. Most of the evidence is for cognitive behavioral therapy, and among the emerging treatments, mindfulness practice and Acceptance and Commitment Therapy offer a suitable and adequate alternative to treat adolescent patients with migraine. These approaches make patients conscious of their problematic suffering condition and able to learn alternative techniques for managing pain and tablet intake, avoiding the risk of evolving into medication overuse headaches and developing any possible side effects of medications [7].

The COVID-19 pandemic was an unprecedented health emergency that affected everyone’s life significantly. It has changed not only routine life but also profoundly altered all clinical activities required to cope with the demand for care of COVID-19 patients. To cope with the care needs of our patients, who have been severely limited due to reduced mobility and the limitations on access to healthcare facilities imposed by law, we had to abruptly change the way we deliver our treatments. We have therefore developed several behavioral treatment programs delivered over the web to educate and support adolescents, and adult patients, suffering from migraine and headaches. So, we were forced to quickly reassess our activity to provide and ensure access to optimal care for all our patients: we implemented telemedicine [12,13] and other web-based modalities, both with the development of new, rigorous, and effective systems to remotely evaluate patients and to deliver therapeutic approaches, including behavioral therapies.

We present the results of a pilot study [14] performed during the COVID-19 emergency, named the Be-Home Kids program. The Be-Home included education on the correct use of drugs, on lifestyle issues, and the delivery of six sessions of a mindfulness-based behavioral approach, by a specific web platform. This modality has demonstrated good effectiveness for the treatment of pain conditions in preceding reports [15].

This paper describes the main results of the 12 month longitudinal course of headache frequency, medication intake, disability, anxiety, depression, and catastrophizing among adolescents attending our program. The general aim of this pilot study was to assess the sustainability and the effect over 12 months on relevant outcomes of our specific web-based protocol, an alternative approach to current practice, particularly reducing face-to-face visits and sessions, taking advantage of facilities offered by new technologies.

## 2. Materials and Methods

### 2.1. The Study and the Intervention

In this mono-center, single-arm, open-label interventional non-pharmacological pilot study, adolescents aged 12–18 with Chronic Migraine (CM) or high-frequency Episodic Migraine (EM) were enrolled to participate in the Be-Home Kids program. Inclusion criteria were: diagnosis of EM without aura (i.e., code 1.1 or the ICHD-3) or CM (i.e., code 1.3 of the ICHD-3) [16]; attendance to secondary or high school during the study period (i.e., the enrolment was carried out in the period September 2020–April 2021 and September 2021–December 2021). Patients were excluded if they had a history of major psychiatric disorders (e.g., personality or psychotic disorders), if they had been on psychotherapy in the previous 18 months, or if they had any previous experience with mindfulness-like approaches. Both patients and caregivers signed an informed consent form prior to enrolment, and the study was approved by the institute’s ethical committee (protocol no. 75.01/2020).

The Be-Home Kids program included education on the correct use of drugs and lifestyle issues and attendance to six sessions of a mindfulness-based behavioral approach, delivered by video calls on a smartphone. Patients participated in the program on a voluntary basis: they were approached on the occasion of the outpatient neurological examination, and those who accepted to participate were given lifestyle indications and instructions for compilation of the research protocol and on how to connect to the online mindfulness sessions.

Patients’ education was focused on lifestyle issues and the correct use of drugs. Patients were therefore instructed to engage in regular physical activity, avoid skipping meals, maintain adequate hydration, avoid or at least limit cigarette smoke, and maintain a regular sleep/wake pattern with at least 7–8 h of sleep per night. With regard to the use of drugs, patients were educated to use medications just in case of a severe pain attack (on a scale from 0 to 5 for the intensity of the attack, when the pain was at least 3).

Six weekly 60 min sessions were scheduled in order to discuss strategies to manage pain and stressful situations that can induce pain episodes and to reinforce the mindfulness practice as previously described [14,17,18]. Sessions were guided by an experienced neurologist (L.G., a leading expert in mindfulness practice for patients with headache disorders) and were held in groups. The intervention was aimed to teach and make direct practice with skills intended to enhance sustained non-judgmental present moment awareness. Different practice experiences were implemented, including guided body scan, tension release, mindfulness meditation, breath-focused imagery, guided imagery, and decentralization of thoughts. Participants were asked to self-practice mindfulness for at least 10 min per day and to practice as much as possible in daily life. For this purpose, they were provided with a 10 min audio record of guided mindfulness meditations. During the sessions, patients were also educated to use breath control, to practice by guided body scan, instructed to tension release, guided imagery, and decentralization of thoughts.

### 2.2. The Research Protocol

The research protocol included measures of headache frequency, our primary endpoint, medication intake, disability, anxiety, depression, and catastrophizing. Headache frequency and medication intake were measured with structured headache diaries and referred to the previous 30 days.

Disability was measured with the Pediatric Migraine Disability Assessment (PedMIDAS) [19]. It is composed of six items referred to the previous 90 days, which address: school days lost due to headaches; school days lost in part due to headaches; functioning at school, i.e., the number of school days in which patients functioned at fewer than half of their abilities due to headaches; days in which home activities, including chores and homework, were not carried out due to headaches; social activity, i.e., days in which patients were not able to participate in play, sports, or social activities due to headaches; days in which patients participated to social activities but at less than 50% of their full ability.

Anxiety was measured with the State-Trait Anxiety Inventory for Children (STAI-C) [20]. It is composed of 40 items, grouped into two subscales (state and trait anxiety), which address the presence of anxiety at the moment in which the questionnaire is filled in, or as a stable trait. Each item is rated on a 1–3 scale describing situations in which patients might have experienced anxiety.

Depression was measured with Kovacs’s Children’s Depression Inventory (CDI) [21]. It is a 27-item scale intended to address and quantify a range of depressive symptoms, including disturbed mood, hedonic capacity, vegetative functions, self-evaluation, and interpersonal behaviors.

Catastrophizing was measured with the Pain Catastrophizing Scale (PCS) [22]. It is composed of 13 items that form three subscales, which identify the three main dimensions of catastrophizing, namely rumination (the constant thinking about pain), magnification (the exaggeration of pain and its consequences), and helplessness (the belief that there is no or limited possibility that pain may improve).

### 2.3. Data Analysis

Descriptive statistics, i.e., frequencies, means, and standard deviations (SD), were used to describe variables.

The analysis was carried out per protocol: for those cases in which at least one follow-up was carried out, we implemented a data imputation using a linear trend at-point approach. With this procedure, each missing value input is determined variable by variable based on the current structure of available data performed, i.e., increasing or decreasing trend from the first subject to the last. Kolmogorov–Smirnov z test was used to test differences between completers and drop-outs.

We addressed changes in primary and secondary endpoints using the non-parametric Friedman test and Wilcoxon test as a post hoc analysis when the Friedman test was significant at *p* < 0.05 level. For descriptive purposes, we also reported changes from baseline for headache frequency and medication intake, using means and 95% Confidence Intervals (95% CI) to represent data. Second, we described the number of patients who reached a reduction in headache frequency higher or equal to 50% compared to baseline.

Data were analyzed using SPSS 27.0.

## 3. Results

A total of 21 patients, of whom three males (14.3%) were included. Table 1 reports baseline features, and the difference between study completers and drop-outs: aged 12–17 (mean 15.2 ± 1.3), and the average disease duration was 3.1 ± 1.7 years (range 1–7). Five patients reported a daily headache, one of them daily intake of symptomatic medications. None of the patients interrupted participation and dropped out during the six weeks of session attendance. Five patients did not complete the study: three dropped out between baseline and month three, one at month six, and one at month 12. Data referred to the last two were imputed, and thus repeated-measure analyses were carried out on 18 cases.

Table 2 reports the longitudinal analyses of study variables. The Friedman test showed a significant time effect showing improvement in headache frequency, medication intake, PedMIDAS, and PCS. The Wilcoxon test, herein used as a post hoc test, showed significant differences between baseline evaluation and each follow-up for headache frequency and PedMIDAS. In addition to this, significant differences between baseline and both three and six-month follow-ups were reported for PCS, whereas no paired comparison showed significance for medication intake.

Figure 1 reports the description of change over time for headache frequency and medication intake from baseline at months 3, 6, and 12. It can be appreciated that the range of variation from baseline is below zero for headache frequency, whereas it always included zero for medication intake.

Over the 12 months, the average reduction in headache frequency was 10.4 ± 10.0 days, corresponding to a percentage reduction of 64 ± 37%. A total of 12 out of 18 who completed the study showed a headache frequency reduction of ≥50%. Among those who did not improve, one showed a stable pattern, and one showed an increase (from 10 to 12 days, i.e., 20% higher).

## 4. Discussion

The paper reports the main result of a mono-center, single-arm, open-label interventional non-pharmacological pilot addressing the effect of the Be-Home Kids protocol in adolescents aged 12–18 with CM or high-frequency EM. The results of the study show that patients report a significant clinical improvement over 12 months, in terms of headache frequency reduction, with a decrease of 10.4 ± 10.0 days, corresponding to around a 64% reduction compared to baseline, medication intake, disability, and pain catastrophizing.

Our results confirm the results of the study by Lovas and colleagues [23], which addressed acceptance of the program and improvements in pain intensity, somatic symptoms, and disability, confirming mindfulness-based intervention as useful in reducing chronic pain conditions. Our results, together with those of Lovas, provide some initial support for the utility of a multimodal approach for young migraineurs, which combines patient education and mindfulness-based programs.

The efficacy of mindfulness treatment is evident from our results on adolescents, but the reason why it can actually have an effect on migraine days has not been unequivocally demonstrated to date. Some authors [7] suggest an indirect action. Indeed the practice of mindfulness acts on the traits of anxiety, depression, and catastrophizing that may play a role in the migraine attack. This is supported by the decrease in all the indicators of these psychological variables indexes recorded at the follow-up in our study.

In addition to these clinical data, it has to be noticed that patients’ adherence during the treatment was very good, as also previously reported by Hesse and colleagues [24]. In fact, none of the patients interrupted the sessions, and we had enough data to run repeated-measure analyses involving 85% of the participants. Considering the overall difficulties that patients and their families had to deal with during the study period (i.e., enrolment in September 2020–April 2021 and September 2021–December 2021), such a result is largely satisfactory. We believe this result was due to the possibility that patients had to take advantage of the treatment directly on their smartphones. However, it is certainly imperative that the patient who is approached for behavioral treatment must be “willing” to such intervention without an attitude of closure and or prejudice.

One of the most prominent aspects of our study was indeed the treatment delivery modality [25]. Online eHealth interventions have been used to address different health outcomes in the clinical population. Among adolescents, these treatment delivery modalities have been shown, in a recent review, to be effective in managing depression, anxiety, stress, and insomnia and improving quality of life when compared with control conditions [26]. Another review, which addressed the effectiveness of web-based interventions used to promote health and behavioral changes in adolescents regarding physical activity, eating habits, tobacco and alcohol use, sexual behavior, and quality of sleep, produced less consistent results [27]. As concluded by the authors, web-based interventions may contribute to health behavior changes among adolescents, but the findings relied on low-quality evidence: thus, large controlled trials with long-term maintenance are needed. More specifically, interventions have shown to be of utility for children and adolescents with chronic illnesses [28] and with pain-related conditions, including headaches [29]. The main problem with the results presented in these reviews is that they are mostly based on relatively small samples, often with heterogeneous populations, and with delivery modalities that are nowadays frankly outdated. For example, as reported in the review by Fisher and colleagues [29], many interventions were delivered by Internet through a computer or relying on CD-ROMs, audiotapes, e-mail, and phone calls. Such modalities are clearly not adequate for the present situation, which is “dominated” by the use of smartphones and their applications, which are more immediate for use by adolescents.

In addition, much of the previous literature is on cognitive behavioral therapy [26,27,28,29], whereas mindfulness-based interventions constituted a minority of the published literature. To the best of our knowledge, the present single-arm study constitutes the first 12 month trial in which a mindfulness-based program was delivered via smartphone to adolescents with CM or high-frequency EM. This experience, which arose from the need to continue to follow our patients involved in mindfulness-based programs during the COVID-19 pandemic, can represent a model of on-demand care to be translated into everyday clinical practice, as also previously pointed out [25]. All clinicians should take advantage of the innovations introduced by technology to reduce the distance between doctors and patients, especially young people, who can often experience disease situations as a real stigma.

A note on participation has to be made. The drop-out rate was 14%, which is in line with our previous experience with mindfulness-based intervention in adolescents (drop-out at the end 11%, the intervention being delivered in person) [11]. The review by Zhou and colleagues [26] reported post-intervention retention rates (i.e., the percentage of patients completing the intervention) for online mental health interventions, mostly targeting anxiety and depression. An analysis of such rates among adolescents and considering therapist-guided and/or smartphone-administered interventions results in rates comprised between 68% and 94%, whereas our rate was 100% at the end of the six weeks. So, we can conclude that our intervention was well accepted by the young patients.

Some limitations have to be acknowledged. First, we did not implement a control condition, such as a “treatment as usual” or waitlist arm. This hinders us from addressing any strict causal relationship between participation in the Be-Home Kids program and clinical improvement. Second, the Be-Home Kids program includes both patients’ education on drug use and lifestyle issues and attendance to mindfulness-based sessions. It is likely that both components had a role in determining variation in clinical parameters over the 12 months, but we cannot disentangle them. Third, we did not specifically record the kind of symptomatic medications patients used to treat acute headaches. We did not encourage the use of any of them as much as possible and suggested using only those that had already proved to be effective. Based on our experience with the group of patients herein described, the used drugs were almost only ibuprofen and paracetamol. Fourth, the sample size was small, drawn from a single center, and composed of two slightly different groups of patients, namely those with EM at high frequency and those with CM. Moreover, three patients dropped out as they decided to stop the treatment and be followed in other headache centers located in their towns. In addition to this, the male:female ratio was slightly unbalanced towards females. However, it should be noted that no sample size determination was implemented. Future RCTs, with adequate sample sizes aimed at comparing attendance mindfulness against a “treatment as usual” condition or as an add-on to the usual condition, are needed.

## 5. Conclusions

In conclusion, our study suggests that an integrated multimodal program such as the Be-Home Kids, implemented during the COVID-19 pandemic through a web-based platform, which combined home-based patient education, and a mindfulness-based practice, produced a sizeable effect on the improvement of headache frequency, medication intake, disability, and pain catastrophizing.

Behavioral therapies are deemed to be free of short- and long-term side effects and may thus be a leading choice in the treatment of migraine in the adolescent population. However, these are struggling to establish themselves as treatment options due to the lack of RCTs and the non-reimbursability by many healthcare systems of these services.

Protocols such as our Be-Home Kids program presented valuable findings in gathering data and experiences on behavioral therapies in headache disorders, especially in the pediatric populations, confirming the feasibility and effect of mindfulness in the treatment of migraine. Based on such a study, RCTs have to be planned and carried out to address efficacy and cost-efficacy issues.

## Figures and Tables

**Figure 1 brainsci-13-00699-f001:**
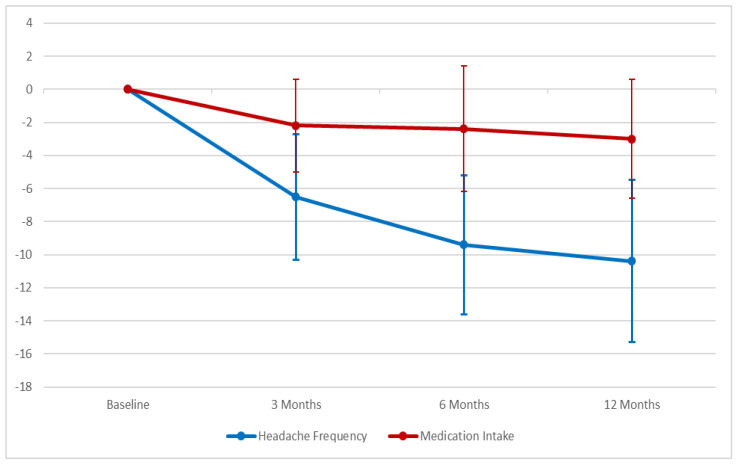
Reduction in headache frequency and medication intake from baseline over 12 months.

**Table 1 brainsci-13-00699-t001:** Baseline features of the whole sample included in the study (N = 21).

	Mean ± SD; N (%)			Completers vs. Drop-Outs (*p*-Value)
	Whole Sample (N = 21)	Study Completers (N = 18)	Study Drop-Outs (N = 3)	
No. Females	18 (86%)	15 (83%)	3 (100%)	0.614
Age	15.2 ± 1.3	15.1 ± 1.3	16.3 ± 0.6	0.203
Migraine Duration	3.1 ± 1.7	3.3 ± 1.7	2.3 ± 1.5	0.832
Headache Frequency	16.0 ± 8.7	14.8 ± 7.9	23.3 ± 11.5	0.541
Medication Intake	5.0 ± 6.5	5.2 ± 6.8	3.7 ± 4.0	1.000
PedMIDAS	67.3 ± 53.6	63.1 ± 49.1	93.0 ± 75.4	0.832
STAI-C1	34.4 ± 5.4	34.6 ± 5.7	33.3 ± 1.6	0.690
STAI-C2	42.1 ± 8.7	41.2 ± 8.8	47.0 ± 4.4	0.541
CDI	11.4 ± 6.7	10.5 ± 6.1	16.3 ± 8.4	0.405
PCS	31.1 ± 10.5	30.9 ± 10.9	32.3 ± 6.1	0.989

Notes. PedMIDAS, Pediatric Migraine Disability Assessment; STAI-C, State-Trait Anxiety Inventory for Children; CDI, Children’s Disability Inventory; PCS, Pain Catastrophizing Scale. Headache Frequency and Medication Intake are monthly. Kolmogorov–Smirnov z test was used to test differences between completers and drop-outs.

**Table 2 brainsci-13-00699-t002:** Per protocol, a longitudinal analysis was carried out on study completers with imputations (N = 18).

	Baseline	3-Month	6-Month	12-Month	Chi-Squared (*p*-Value)	Wilcoxon Post-Hoc Z (*p*-Value)
Headache Frequency	14.8 ± 7.9	8.3 ± 8.9	5.3 ± 5.3	4.3 ± 5.2	22.0 (*p* < 0.001)	Baseline-3M: Z = −2.77 (*p* = 0.006) Baseline-6M: Z = −3.44 (*p* < 0.001) Baseline-12M: Z = −3.56 (*p* < 0.001)
Medication Intake	5.2 ± 6.8	3.0 ± 4.2	2.8 ± 3.0	2.2 ± 3.3	9.6 (*p* = 0.023)	No post hoc significance at *p* < 0.05 level
PedMIDAS	63.1 ± 49.1	35.3 ± 38.7	39.5 ± 41.9	24.6 ± 20.9	12.5 (*p* = 0.006)	Baseline-3M: Z = −2.35 (*p* = 0.019) Baseline-6M: Z = −2.15 (*p* = 0.031) Baseline-12M: Z = −3.05 (*p* = 0.002)
STAI-C1	34.6 ± 5.7	35.4 ± 7.0	32.9 ± 6.9	33.4 ± 5.6	1.4 (*p* = 0.698)	–
STAI-C2	41.2 ± 8.8	39.6 ± 8.8	36.3 ± 6.8	37.0 ± 7.1	5.9 (*p* = 0.117)	–
CDI	10.5 ± 6.1	8.6 ± 4.3	7.9 ± 5.4	10.2 ± 7.6	4.8 (*p* = 0.118)	–
PCS	30.9 ± 10.9	25.7 ± 9.4	23.8 ± 10.4	25.6 ± 10.2	9.6 (*p* = 0.022)	Baseline-3M: Z = −2.25 (*p* = 0.025) Baseline-6M: Z = −2.77 (*p* = 0.006)

Notes. PedMIDAS, Pediatric Migraine Disability Assessment; STAI-C, State-Trait Anxiety Inventory for Children; CDI, Children’s Disability Inventory; PCS, Pain Catastrophizing Scale. Headache Frequency and Medication Intake are monthly.

## Data Availability

Data ara available from corresponding author upon request.

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
