# Peer review of "The Be-Home Kids Program: An Integrated Approach for Delivering Behavioral Therapies to Adolescents with Episodic and Chronic Migraine"

_brainsci, 2023, doi:10.3390/brainsci13040699_

Round 1

Reviewer 1 Report

This is an open study evaluating an online programme for headache prevention in children. The presentation of the rationale, methodology and results is appropriate. The discussion adequately relates the results to the literature. As a pilot study, this work is interesting.

Author Response

Thanks for your comment. We agree about your observation. 

Reviewer 2 Report

Thank you for this interesting submission. This is important work!

The abstract needs some changes. There ARE specific indications for pharmacologic treatments for adolescent migraine (ie- AAN guidelines). Additionally, it is incorrect to say that behavioral approaches are totally free from side effects. It is well known that psychological therapies can uncover distressing thoughts and cause distress and in some cases, worsening mood/temporarily worsening mental health. This should be closely monitored by a professional. It is never correct to say that a treatment (pharmacological or non-pharmacological) is free from potential adverse effects.

Additionally, lines 21 and 54 and 55 have misspelled words. Additionally, I found at least three other typos while reading.

Is "BE-HOME" an acronym? If not, why are all of the letters capitalized? This is confusing.

Line 110- you should define CM and EM (what the abbreviations stand for) the first time that you use them in the manuscript.

I don't know what "None of the patients interrupted participation during the sessions" means. Of the participants who completed the study, did all of them complete all 6 video sessions?

In the discussion you should include mention of your significant drop-out rate, and how this compares to other studies. This needs to be mentioned as a limitation.

In the discussion, please include a discussion of how your results compare to the natural history of pediatric migraine, since migraine tends to get better over time even with no treatment.

Author Response

Thanks for your comments. 

1) The abstract needs some changes. There ARE specific indications for pharmacologic treatments for adolescent migraine (ie- AAN guidelines). Additionally, it is incorrect to say that behavioral approaches are totally free from side effects. It is well known that psychological therapies can uncover distressing thoughts and cause distress and in some cases, worsening mood/temporarily worsening mental health. This should be closely monitored by a professional. It is never correct to say that a treatment (pharmacological or non-pharmacological) is free from potential adverse effects.

R1) We would say that there are no “specific indications” for the pharmacologic treatment of pediatric migraine that provide solid background. We went through the ANN “Practice guideline update: acute treatment of migraine in children and adolescents” (August 2019), as well as through the ANN “Practice guideline update summary: Pharmacologic treatment for pediatric migraine prevention” (September 2019). Both the two guidelines consistently state that “there is insufficient evidence” for the use of any product with regard to efficacy issues. All the recommendations with regard to medications’ use are Level B or C.

With regard to the second part of your comment, for sure side effects can be present in behavioral approaches. However, it is important to distinguish a “psychotherapy” of any approach (psychodynamic, systemic, behavioral, cognitive and so on) from a “behavioral intervention” like the one we implemented. Our intervention focuses on the mastery of few techniques directed to headache management through increasing awareness of inner states. Clearly, this might “move” something inside participants, but we would not say that the degree of experiencing “worsening mood/temporarily worsening mental health” is somehow similar to what can be observed among patients attending a psychotherapy or any approach. We think much of this confusion is due with the way it was written before, so we rephrased it and wrote that “they are generally associated to less or no unpleasant effects” both in the abstract (lines 17-18) and in introduction (line 77).

2) Additionally, lines 21 and 54 and 55 have misspelled words. Additionally, I found at least three other typos while reading.

R2) We modified the line 21 sentence into “It was delivered by web and included education on drugs’ use, lifestyle issues, and six sessions of mindfulness-based behavioral practice.” so to enhance clarity. On lines 54-55, we found no misspelled words.

3) Is "BE-HOME" an acronym? If not, why are all of the letters capitalized? This is confusing.

R3) It is not really an acronym: the intervention was given this name in consideration of the “stay at home” slogan used during the pandemic. We amended the use of capital letters to avoid confusion.

4) Line 110- you should define CM and EM (what the abbreviations stand for) the first time that you use them in the manuscript.

R4) EM and CM stand for episodic and chronic migraine. We specified (lines 111-112)

5) I don't know what "None of the patients interrupted participation during the sessions" means. Of the participants who completed the study, did all of them complete all 6 video sessions?

R5) It means that none of the patients dropped out during the 6 weeks of session attendance. We rephrased as “None of the patients interrupted participation and dropped out during the six weeks of sessions’ attendance”, see lines 199-200.

6) In the discussion you should include mention of your significant drop-out rate, and how this compares to other studies. This needs to be mentioned as a limitation.

R6) 3 out of 21 is 14%. Considering the period in which the study was carried out, we would not say this rate is “significant”. In another study on pediatric mindfulness delivered in person (reference 11), we had 11% attrition. The review we mentioned on online mental health interventions (ref 26) reports as common indicator the retention rate post intervention and, for the population of adolescents and therapist-guided interventions or smartphone app-based ones, the percentages range between 68% and 94%, whereas in the group of patients herein described the retention rate was 100% (as none of the patients interrupted participation during the sessions). So, in discussion we wrote that the drop-out rates we observed were similar to those reported in available literature and higher considering post-intervention retention rates, see lines 293-301 [A note on participation has to be made ... well accepted by the young patients.]

7) In the discussion, please include a discussion of how your results compare to the natural history of pediatric migraine, since migraine tends to get better over time even with no treatment.

R7) Migraine tends to get better over time even with no treatment in part of the cases. The estimates from the GBD study show that migraine prevalence and disability tend to increase up to the age of 50 and decline thereafter. Parallel to this, there are data that show a positive clinical evolution of migraine in this category of patients: nevertheless, in our patients the condition of chronic migraine and high frequency of migraine enforces us to propose a non-invasive preventive treatment to teach patients to manage migraine episodes avoiding as much as possible the use of medications. Our patients were in a bad conditions and their migraine evolution was not positive. No change was made to the manuscript.

Reviewer 3 Report

I am not clear about certain aspects of this study.

My main concern is that there has been no matched, randomly selected, control group that permits taking into account the variability of the course of the natural history of migraine, treated or untreated, over a period of a year. As I see it, the contents of the paper show that your approach can be delivered reasonably successfully, but I am less convinced about the degree of patient benefit that it has, of itself, produced. I note your statement that the approach would not cause adverse effects, but did you actually investigate this or assume it?

Was your analysis on an intention-to-treat basis, as the contents of table 1 suggests, or on the basis of the treatment course completed, as the text seems to suggest? If on an intention-to-treat basis, how were the headache and medication frequencies of the drop-outs handled?

I could see no indication as to whether any pharmacological prophylactic measures were involved or what the natures of the medications used were. There are old publications indicating that regular use of aspirin or ergotamine could help prevent migraine recurrences, and frequent intermittent use might do likewise.

Are you recording frequency of individual migraine attacks, frequency of any sort of headache episode, or numbers of headache (or migraine) affected days per month. I suspect is probably the latter, but if so at baseline on the average medication was used on only roughly 5 of roughly 15 headache-affected days, suggesting that much of the headache must have been quite mild. This interpretation may correlate with the data of figure 1, where the reduction in headache frequency seems greater than the reduction in medication use.

The 18:3 female to male preponderance at outset seems higher than I would have anticipated. The gender of the dropouts is not stated, and it could even be that the paper’s findings really deal only with adolescent females if all the males dropped out.

Author Response

Thanks for your comments. 

1) My main concern is that there has been no matched, randomly selected, control group that permits taking into account the variability of the course of the natural history of migraine, treated or untreated, over a period of a year. As I see it, the contents of the paper show that your approach can be delivered reasonably successfully, but I am less convinced about the degree of patient benefit that it has, of itself, produced. I note your statement that the approach would not cause adverse effects, but did you actually investigate this or assume it?

R1) We understand your concern, but the paper is to be taken for what it is: a single arm pilot study, aimed to address the sustainability of this approach. We actually misused the second term, i.e. effectiveness: the use of effect is more correct as no comparator treatment was implemented in a kind of control group, so we updated it (line 104).

2) Was your analysis on an intention-to-treat basis, as the contents of table 1 suggests, or on the basis of the treatment course completed, as the text seems to suggest? If on an intention-to-treat basis, how were the headache and medication frequencies of the drop-outs handled?

R2) No, the analysis is per protocol with the imputation of missing data when at least one follow-up was available in the record set for each patient, as written on line 181-183. We understand the confusion in reason of the lack of comparison between drop-outs and completers (after imputation): we run the non-parametric Kolmogorov-Smirnov z test (see lines 185-186 in methods) which is suitable for small samples and added columns in table 1, and a small text on line 196.

3) I could see no indication as to whether any pharmacological prophylactic measures were involved or what the natures of the medications used were. There are old publications indicating that regular use of aspirin or ergotamine could help prevent migraine recurrences, and frequent intermittent use might do likewise.

R3) Thanks for asking this: all patients included into the study were not treated by pharmacological therapy for prevention of migraine and were educated and instructed to avoid symptomatic medications, specifically NSAIDs, as much as possible. We did not encourage the use of aspirin or ergotamine, also in reason of the fact that the latter is not allowed in Italy for pediatric populations. We did not record the kind of drugs, but these patients consumed almost only ibuprofen and paracetamol. A comment on this was added to the limitations, see lines 308-313 [Third, we did not specifically record the kind of symptomatic medications that patients used to treat acute headaches. We did not encourage the use of any of them as much as possible, and suggested to use only those that already proved to be effective. Based on our experience with the group of patients herein described, the used drugs were almost only ibuprofen and paracetamol.]

4) Are you recording frequency of individual migraine attacks, frequency of any sort of headache episode, or numbers of headache (or migraine) affected days per month. I suspect is probably the latter, but if so at baseline on the average medication was used on only roughly 5 of roughly 15 headache-affected days, suggesting that much of the headache must have been quite mild. This interpretation may correlate with the data of figure 1, where the reduction in headache frequency seems greater than the reduction in medication use.

R4) We addressed headache frequency among patients diagnosed with episodic and chronic migraine. This does not imply that all headaches had migraine features as per ICDH-3 diagnostic criteria. In this context, it has to be noted that enrolled patients were not using so many medications at the beginning of the program: however, this does not imply that pain intensity was always “quite mild”. We can only reasonably presume that a portion of headaches was more severe than the remaining part of the attacks: however, we do not have a specific indication on this, so it would be quite speculative to hypothesize whether these attacks were more intense due to the fact that they had migraine-like symptoms or in reason of association with any other trigger, including sport-related, school-related, rather than menstrual-related ones. In any case, patients received instructions to avoid medications as much as possible: this is the most reasonable explanation for the large decrease in headache days, compared to the small one in medication intake, i.e. the fact that medication intake was quite low at baseline and therefore there was clearly less room for reduction. No change was made to the manuscript.

5) The 18:3 female to male preponderance at outset seems higher than I would have anticipated. The gender of the dropouts is not stated, and it could even be that the paper’s findings really deal only with adolescent females if all the males dropped out.

R5) The three who dropped out were all females, as show in the updated table 1. So, the male:female ratio is 1:6 considering all patients and 1:5 considering those used for the analyses. The expectable ratio should be something like 1:4 to 1:5. However, as stated before, this is to be intended as a pilot study: we did not implement any sample size calculation, so we can presume it is largely casual. We restructured the last limitation to include all issues related to sample, see lines 313-318 [Fourth, the sample size was small, drawn from a single center, and composed of two slightly different groups of patients, namely those with EM at high frequency and those with CM. Moreover, three patients dropped-out as they decided to stop the treatment and to be followed in other headache centers located in their towns. In addition to this, the male:female ratio was slightly unbalanced towards females: it has however to be noted that no sample size determination was implemented.]

Round 2

Reviewer 3 Report

I have no further matters to raise